# Amodal SAM: Open-World Amodal Segmentation

## Abstract

Amodal segmentation, which aims to predict complete object shapes including occluded regions, remains challenging in open-world scenarios, where models must generalize to novel objects and contexts. While the Segment Anything Model (SAM) has demonstrated remarkable zero-shot generalization capabilities, it is fundamentally limited to visible region segmentation. This paper presents Amodal SAM, a framework that extends SAM's capabilities to amodal segmentation while preserving its powerful generalization ability. The improvements lie in three aspects: (1) a lightweight Spatial Completion Adapter that enables occluded region reconstruction, (2) a Target-Aware Occlusion Synthesis (TAOS) pipeline that addresses the scarcity of amodal annotations by generating diverse synthetic training data, and (3) novel learning objectives that enforce regional consistency and topological regularization. Extensive experiments demonstrate that Amodal SAM achieves state-of-the-art performance on standard benchmarks while exhibiting strong generalization to novel scenarios. Furthermore, our framework seamlessly extends to video sequences, as the first attempt to tackle the open-world video amodal segmentation. We hope that our research can advance the field toward practical amodal segmentation systems that can operate effectively in unconstrained real-world environments. Code will be made publicly available.

## 1 Introduction

The human visual system possesses the ability to interpolate unseen information, particularly through amodal perception, our capacity to mentally complete partially occluded objects. This innate ability has inspired a computer vision task called amodal segmentation, which aims to predict complete object shapes, including their hidden portions.

Nonetheless, amodal segmentation remains challenging, as the existing methods primarily focus on amodal segmentation within the same domain witnessed during training. However, in open-world scenarios, models must generalize to novel object categories and contexts that have not been seen during model training, which poses challenges to current amodal segmentation models, as shown in Figure 1.

To enhance open-world visual perception capabilities, the Segment Anything Model (SAM) was introduced, demonstrating remarkable generalization to new samples. However, SAM is inherently constrained to segment visible regions, thereby lacking the capacity to address

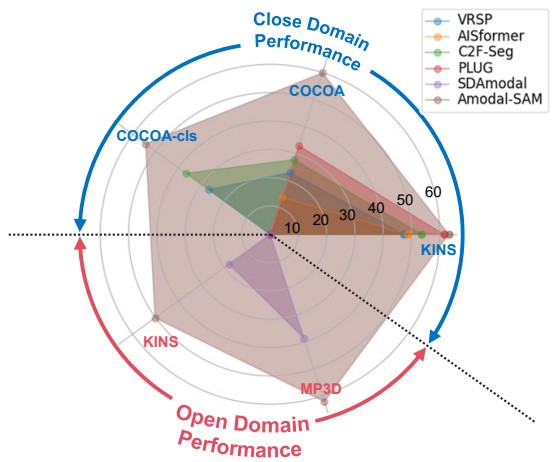

Figure 1: This figure shows the comparisons between our Amodal SAM and previous state-of-the-art models in the closed and open domains. In both settings, using $mIoU_o$ as the evaluation metric, our method outperforms the state-of-the-art model on all compared datasets.

open-world challenges in amodal segmentation. This raises the question: *Is it viable to effectively adapt the SAM model for amodal segmentation while preserving its generalization prowess?*

Therefore, in this work, we present Amodal SAM, a framework that extends SAM's capabilities to amodal segmentation while preserving its powerful zero-shot ability. Our key insight is that successful adaptation requires a holistic approach addressing three complementary aspects: model architecture, training data, and optimization objectives. Specifically, for model adaptation, we introduce a lightweight Spatial Completion Adapter (SCA) that enables the model to reconstruct occluded regions while maintaining SAM's core segmentation abilities. To overcome the scarcity of large-scale amodal annotations required by open-world training, we propose Target-Aware Occlusion Synthesis (TAOS), an efficient pipeline that synthesizes diverse occlusion patterns from existing segmentation datasets without manual labeling. Finally, we introduce the learning objectives regarding regional consistency and holistic topological regularization to facilitate training.

The effectiveness of the proposed Amodal SAM is demonstrated through extensive experiments across extensive datasets on both challenging image and video benchmarks. Amodal SAM not only achieves state-of-the-art performance on standard amodal segmentation tasks but also shows strong generalization to novel object categories and scenes. Furthermore, we show that our framework can be seamlessly extended to video amodal segmentation by adapting SAM-2, highlighting its flexibility and broad applicability. To the best of our knowledge, it is the first attempt to address video amodal segmentation in the open-world settings.

To summarize, our contributions are as follows:

- We observe that existing amodal segmentation models lack open-world generalization capabilities, significantly limiting their real-world applications.

- With the improvements across three dimensions - model, data, and optimization - we successfully adapt SAM, a generic foundation model, to amodal segmentation, allowing the model to generalize to novel and diverse open-world scenarios.

- The obtained framework, *i.e.*, Amodal SAM, achieves superior performance in both closed and open-world scenarios, and can be easily extended to video applications, marking the first successful implementation of open-world video amodal segmentation.

## 2 PRELIMINARIES

### 2.1 AMODAL OBJECT SEGMENTATION

**Task description.** Amodal object segmentation (Zhu et al.; Li & Malik; Nguyen & Todorovic; Tran et al., a;b; Qi et al.; Zhan et al., a) aims to predict the complete pixel-level mask of a target object that is partially obscured. Formally, according to (Qi et al.; Tran et al., b;a), a rough bounding box $B$ that covers the region-of-interest (ROI) in a specific image $I$ will be provided, to identify both the visible and occluded regions of the target object through the prediction mask $M$. With the amodal segmentation model $\mathcal{F}$, this procedure can be formulated as:

$$M = \mathcal{F}(B, I). \tag{1}$$

Certain approaches (Qi et al.; Li & Malik; Follmann et al.; Zhang et al., 2019) may use conventional segmentation algorithms, such as (Ronneberger et al.; Long et al.; He et al.), to explicitly derive the visible region mask from the region-of-interest (ROI) box $B$ corresponding to the target object as an input component. In this case, the formulation becomes:

$$M = \mathcal{F}(s(B), I), \tag{2}$$

where $s()$ denotes the conventional segmentation algorithm that transforms the visible part within the ROI box $B$ into a mask. For both Eqs. equation 1 and equation 2, the ground truth mask of the target object is provided to supervise the predicted mask $M$ during training.

**Issues in the open-world scenarios.** In the existing literature, amodal segmentation is typically carried out within a predetermined set of target object classes (Zhu et al.; Qi et al.; Follmann et al.). This constrained setting limits the model's ability to handle novel object categories in real-world scenarios. Furthermore, we have noticed that despite having the same categories, current amodal

segmentation models struggle to address the domain shift across various tasks effectively. For instance, a model trained on indoor scenes might struggle with outdoor environments.

Hence, creating a resilient amodal segmentation framework capable of generalizing to unforeseen scenarios is crucial for practical real-world applications. In this work, we tackle this challenge by leveraging SAM's (Kirillov et al.) strong generalization capability.

## 2.2 SEGMENT ANYTHING MODEL (SAM)

SAM (Segment Anything Model) (Kirillov et al.), developed by Meta AI, is a foundation model for image segmentation with three key components: 1) an image encoder that extracts dense visual features; 2) a prompt encoder that processes various forms of user inputs; and 3) a mask decoder that synthesizes these signals to generate precise segmentation masks. The remarkable generalization capability of SAM has inspired numerous downstream adaptations (Chen et al., 2023; Li et al., 2023; Ma & Wang, 2023; Ke et al.). Drawing inspiration from the adapter paradigm originally proposed in natural language processing(Houlsby et al.; Hu et al.), recent works such as SAM-Adapter(Chen et al., 2023) and ViT-Adapter(Chen et al.) have demonstrated the effectiveness of lightweight adapter modules in transferring knowledge from ViT-based architectures to specific domains.

While we follow the principle of adapter-based tuning to extend SAM's capabilities to amodal segmentation, we observe that the naive adapter-based adaptation schemes yield suboptimal performance for this challenging task as shown in later experiments.

## 3 AMODAL SAM

### 3.1 OVERVIEW

As amodal segmentation in the open world requires robust zero-shot generalization to handle novel object classes and data distributions, in this work, we propose Amodal SAM - a framework that decently extends SAM's capabilities from visible region segmentation to amodal segmentation while preserving SAM's strong zero-shot capabilities. The proposed paradigm focuses on the training phase about the following three synergistic aspects:

- *Model adaptation* enables efficient adaptation with minimal structural modification through the lightweight gated adapter.
- *Data adaptation* facilitates the supervised tuning via synthesizing object occlusions without the need for exhaustive human annotation.
- *Optimization adaptation* involves transitioning from a generic vision foundation model to a specialized amodal segmentation model by incorporating specific learning objectives.

In the following, Sections 3.3, 3.2, and 3.4 will detail our adaptation strategies for model architecture, training data, and optimization processes, respectively. Subsequently, Sec. 3.5 demonstrates how our designs can be easily extended to video amodal segmentation with SAM-2(Ravi et al., 2025), highlighting the generalization capabilities.

### 3.2 MODEL ADAPTATION

In this study, we investigate adapters for model adaptation to enhance the capacity to perceive the occluded region.

**Encoder-focused adaptation.** Instead of inserting the adapters into both the encoder and decoder of SAM, we adopt an encoder-focused adaptation strategy.

Specifically, the SAM encoder handles feature extraction, while the decoder primarily converts features into masks. Consequently, the encoder may encounter a more substantial domain disparity when faced with different tasks. If decoder tuning were implemented, it might jeopardize SAM's inherent mask generation abilities, as later demonstrated in Section 4.4. Therefore, an encoder-focused adaptation can bridge the domain gap across diverse tasks while preserving the model's core functionality with the frozen decoder. The overall modeling process of Amodal SAM can be

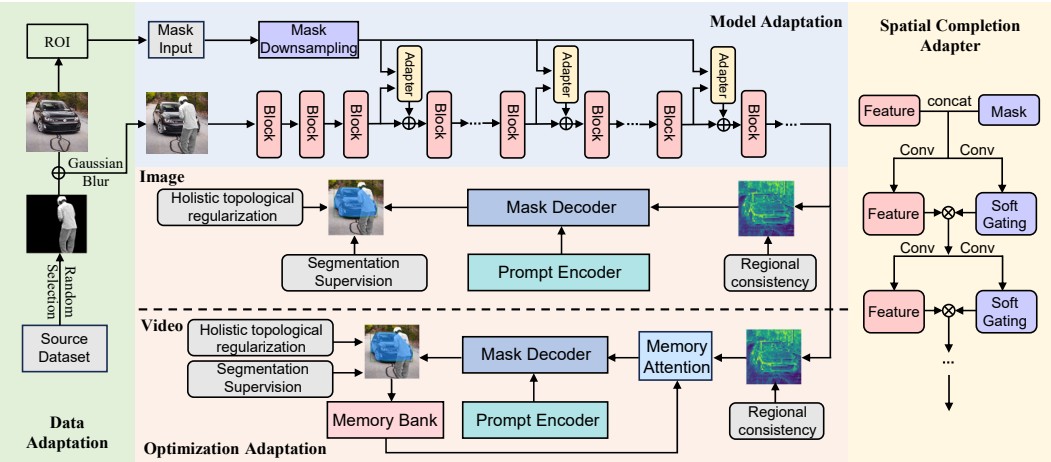

Figure 2: The overall structure of the proposed Amodal Spatial Attention Module (SAM) includes three key aspects for the adaptation from SAM to Amodal SAM: data, model, and optimization. The process of data adaptation involves an automated pipeline for converting generic segmentation annotations into those necessary for amodal segmentation. Model adaptation is realized through the Spatial Completion Adapter (SCA) in a manner focused on the encoder-tuning. Optimization adaptation is performed using a composite learning objective.

expressed as:

$$M = \mathcal{F}_{\text{dec}}(\mathcal{F}_{\text{prompt}}(B), \mathcal{F}'_{\text{a-enc}}(I)), \tag{3}$$

where $M$ is the predicted mask for both visible and occluded regions of the target object, which is specified by an ROI box $B$ on the input image $I$. $\mathcal{F}_{\text{dec}}$ and $\mathcal{F}_{\text{prompt}}$ denote SAM's mask decoder and prompt encoder respectively, while $\mathcal{F}'_{\text{a-enc}}$ represents the adapted SAM encoder tailored for amodal segmentation.

**Prior-guided feature extraction.** For the amodal segmentation task, an ROI box $B$ is needed in the input to indicate the target. We found it is beneficial to additionally incorporate a binary prior mask $M_{\text{spec}}$ into the feature encoder, as the spatial prior to guide the target-related feature generation. During inference, $M_{\text{spec}}$ can be obtained from $B$ by setting the value to 1 for the regions inside $B$ and to 0 for the remaining regions of the binary mask $M_{\text{spec}}$. To this end, Eq. equation 3 is accordingly updated as:

$$M = \mathcal{F}_{\text{dec}}(\mathcal{F}_{\text{prompt}}(B), \mathcal{F}'_{\text{a-enc}}(I, M_{\text{spec}})). \tag{4}$$

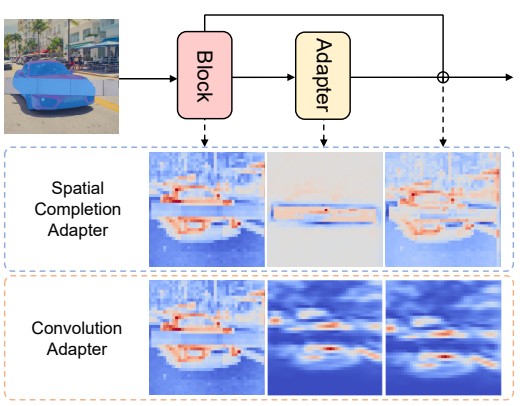

Figure 3: The figure demonstrates the effects of the Spatial Completion Adapter (SCA). It is evident that SCA effectively complements the input by restoring features in the occluded regions.

It is important to note that the mask $M_{\text{spec}}$ might not be accessible during model training, given the absence of a dataset containing both occluded and full segmentation ground-truth masks of the target object. Our approach to tackle this challenge will be presented in Section 3.3.

**Spatial Completion Adapter (SCA).** In Eq. equation 4, incorporating the spatial prior via $M_{\text{spec}}$ into the encoder underscores the need to devise a dedicated encoder capable of accommodating this guided enhancement. Hence, we propose the Spatial Completion Adapter (SCA) and integrate it into the baseline SAM encoder. SCA is designed to reconstruct obscured regions of target objects within the feature space, leveraging the spatial cues offered by $M_{\text{spec}}$, thereby facilitating the completion of occluded areas.

Unlike conventional adapters (Li et al.; Yuan et al.; Ranftl et al.) that rely on basic convolutions or linear transformations for cross-domain feature adaptation, the proposed SCA draws inspiration from Gated Convolution (Yu et al.). By incorporating spatial soft gating in the feature space, SCA enables dynamic feature selection that allows SCA to effectively utilize spatial cues from $M_{\text{spec}}$ and reconstruct occluded regions. In other words, for the completion of occluded areas, SCA will pay more attention to the regions highlighted by $M_{\text{spec}}$, as shown in Figure 3, effectively recovering the information in the occluded region.

Specifically, as illustrated in Fig. 2, SCA acquires element-wise gating weights $\mathbf{G}$ from the feature map $\mathbf{E} \in \mathbb{R}^{W \times H \times C}$ and spatial guidance $M_{\text{spec}}$ to derive spatially-informed features $\mathbf{O}$. This operation can be expressed as:

$$
\begin{aligned}
\mathbf{G} &= \sigma(\mathcal{F}_{\text{gate}}(\mathbf{E}, M_{\text{spec}})), \\
\mathbf{O} &= \mathbf{G} \odot \phi(\mathcal{F}_{\text{feat}}(\mathbf{E}, M_{\text{spec}})).
\end{aligned}
\tag{5}
$$

Here, $\mathbf{G} \in \mathbb{R}^{W \times H \times C}$ and $\mathbf{O} \in \mathbb{R}^{W \times H \times C}$ represent the learned gating weights and filtered features, respectively. $\mathcal{F}_{\text{gate}}$ and $\mathcal{F}_{\text{feat}}$ are transformation functions implemented using vanilla convolutional layers. $\sigma$ and $\phi$ denote the Sigmoid and LeakyReLU (Şenyiğit et al., 2014) activation functions, respectively. More structural details regarding SCA are available in the supplementary material.

### 3.3 DATA ADAPTATION

Amodal segmentation poses a fundamental challenge that sets it apart from traditional object segmentation by necessitating precise delineation of obscured object regions.

However, as the data scale is essential to achieve an open-world model, existing amodal segmentation datasets (Zhu et al.; Follmann et al.; Qi et al.) are limited in scale and scope, typically covering only specific domains with restricted object categories, making them inadequate for open-world applications. Moreover, manually curating a large-scale dataset with sufficient amodal masks for occluded objects would be prohibitively time-consuming and expensive.

To address these challenges, we introduce a Target-Aware Occlusion Synthesis (TAOS) pipeline to accomplish the data adaptation. The TAOS pipeline can efficiently convert standard segmentation annotations available in the large-scale SA-1B dataset (Kirillov et al.) into the required formats for amodal training, eliminating the necessity for manual labeling. Further details are outlined below.

**Target-Aware Occlusion Synthesis (TAOS).** Since the current extensive segmentation dataset only labels the visible regions, within TAOS, we choose to artificially create and model occlusions that could potentially occur among distinct objects.

To generate an image featuring an occluded object, we first randomly select an image from the original dataset, *i.e.*, SA-1B, containing an object within a predefined size range, designating this object as the *target*. Subsequently, we randomly crop an object (or a portion thereof) from another randomly selected image, ensuring that it is of a suitable size relative to the target object, to serve as the *occluder*. Then, we overlay the *occluder* onto the *target* object at a random position and with a random overlapping area within a specified range. Furthermore, we employ VLM to evaluate the generated occlusion and eliminate invalid data.

Finally, we apply pixel blurring to the boundaries of the occluded regions to ensure the naturally synthesized occlusion. Specifically, we adopt Gaussian Blur, which smooths each boundary pixel by normalizing the pixel value according to a predefined Gaussian kernel. Formally, a boundary pixel located at $(x, y)$ in an image $I$ is smoothed as follows:

$$
\begin{aligned}
G(x, y) &= \frac{1}{2\pi\sigma^2} \exp\left(-\frac{x^2 + y^2}{2\sigma^2}\right), \\
I'(i, j) &= \sum_{x=-\frac{k}{2}}^{\frac{k}{2}} \sum_{y=-\frac{k}{2}}^{\frac{k}{2}} G(x, y) \cdot I(i + x, j + y).
\end{aligned}
\tag{6}
$$

Here, $G$ denotes the Gaussian kernel with size $k$. The overall pipeline is illustrated in Figure 4.

Figure 4: The illustration of the proposed Target-Aware Occlusion Synthesis (TAOS) pipeline. Initially, by randomly selecting masks and superimposing them on the target object, we create an amodal mask, a visible mask, and an occlusion mask. Furthermore, applying Gaussian blur to the edges helps in effectively smoothing the transition between the two objects. Details can be found in Section 3.3.

## 3.4 OPTIMIZATION ADAPTATION

The model architecture and training data adaptations necessitate refining SAM's optimization process to better suit amodal segmentation requirements. In addition to the vanilla segmentation loss $\mathcal{L}_{\text{seg}}$, we introduce two additional learning objectives for amodal segmentation: (1) regional consistency $\mathcal{L}_{\text{reg}}$ between the visible and occluded regions of the target object, and (2) holistic topological regularization $\mathcal{L}_{\text{hol}}$ based on adversarial learning. To this end, the overall learning objective $\mathcal{L}$ is formulated as:

$$\mathcal{L} = \mathcal{L}_{\text{seg}} + \mathcal{L}_{\text{reg}} + \mathcal{L}_{\text{hol}}. \tag{7}$$

**Segmentation Supervision.** To begin with, we first adopt Dice loss and BCE loss to optimize the segmentation prediction:

$$\mathcal{L}_{\text{seg}} = \mathcal{L}_{\text{Dice}} + \alpha \mathcal{L}_{\text{BCE}}, \tag{8}$$

Where $\alpha$ is a balancing factor that is set to 10, following (Cheng et al.; Ma & Wang, 2023; Chen et al., 2023).

**Regional consistency.** Then, based on the rationale that visible and occluded regions of the same object are expected to exhibit similar characteristics like appearance and texture patterns, we incorporate intra-object regional consistency during training. This is achieved by minimizing the difference between the representations of visible and occluded regions:

$$\mathbf{E}_{\text{vis}} = \frac{\sum \mathbf{E} \odot \mathbf{M}_{\text{vis}}}{\sum \mathbf{M}_{\text{vis}}}, \quad \mathbf{E}_{\text{occ}} = \frac{\sum \mathbf{E} \odot \mathbf{M}_{\text{occ}}}{\sum \mathbf{M}_{\text{occ}}},$$
$$\mathcal{L}_{\text{reg}} = 1 - \cos(\mathbf{E}_{\text{vis}}, \mathbf{E}_{\text{occ}}). \tag{9}$$

Here, $\mathbf{E} \in \mathbb{R}^{W \times H \times C}$ denotes the feature map extracted by the adapted encoder $\mathcal{F}'_{\text{a-enc}}(I, M_{\text{spec}})$. With the visible mask $\mathbf{M}_{\text{vis}} = 1 - \mathbf{M}_{\text{occ}}$ and the occlusion mask $\mathbf{M}_{\text{occ}}$, we can obtain $\mathbf{E}_{\text{vis}} \in \mathbb{R}^C$ and $\mathbf{E}_{\text{occ}} \in \mathbb{R}^C$, indicating the representations of the visible and occluded regions, respectively. The function $\cos()$ calculates the cosine similarity between the inputs. Therefore, by minimizing the regional consistency loss $\mathcal{L}_{\text{reg}}$, the visible and occluded regions are encouraged to have more similar representations.

**Holistic topological regularization.** Furthermore, we introduce a topological regularization that preserves the high-order structural relationships between the predicted mask and ground truth, enabling the explicit learning of shape priors through adversarial training. Specifically, we implement a discriminator $D$ that evaluates the topological and morphological consistency between the predicted mask and ground truth. The optimization objective for the discriminator $D$ is formulated as:

$$\mathcal{L}_{\text{hol}} = \min_{\text{A-SAM}} \max_{D} (\mathbb{E}_{M_g \sim P_{\text{GT}}(M)} [\log D(M_g, I)] + \mathbb{E}_{M_p \sim P_{\text{A-SAM}}(M)} [\log(1 - D(M_p, I))]), \tag{10}$$

where $P_{\text{GT}}$ and $P_{\text{A-SAM}}$ are the distributions of ground truth masks and predicted masks by the proposed Amodal SAM, respectively. More details, such as the discriminator's structure, are in the appendix A.1.2.

## 3.5 VIDEO AMODAL SEGMENTATION

The recently introduced SAM-2 model demonstrates enhanced capabilities in segmenting diverse elements within video data, representing a substantial advancement over its predecessor. This development enables the seamless application of our proposed adaptation strategies—encompassing model architecture, data processing, and optimization techniques—to video segmentation tasks using the SAM-2 framework. This successful extension not only validates the generalization capability of our proposed designs but also significantly advances the field of open-world amodal video segmentation.

The structural consistency between SAM2 and SAM enables straightforward extension of the proposed approach to video amodal segmentation. Akin to the approach used in the original SAM model, we can leverage adapter tuning to refine the encoder of SAM-2 while maintaining the other components fixed. In contrast to SAM's Vision Transformer (ViT)-based encoder, SAM-2 integrates Hiera (Ryali et al.) that is specifically tailored for video tasks. Hiera comprises four feature encoding stages that enable the learning of multi-scale features. To adapt each scale of features individually, we incorporate the proposed Spatial Completion Adapter (SCA) into each encoding stage.

Unlike image amodal segmentation, video amodal segmentation faces unique challenges due to temporal variations in object occlusion patterns and positional changes across frames, making it infeasible to process video frames in $M_{\text{spec}}$ using the same frame-by-frame approach as with images. To address this issue, we leverage the spatiotemporal correlations of object positions between adjacent frames in videos. By using the prediction results from the previous frame as the input adapter for $M_{\text{spec}}$ in the current frame, we establish a progressive temporal approach. Specifically, we utilize the prediction output of each frame to create a coarse region of interest (ROI) for the $M_{\text{spec}}$ input of subsequent frames. Further implementation details of the proposed video amodal segmentation can be found in the appendix A.1.4.

## 4 EXPERIMENTS

### 4.1 EXPERIMENTAL SETUP

**Data sources of TAOS pipeline.** The SA-1B dataset comprises 11 million varied, high-resolution, privacy-protecting images and 1.1 billion top-notch segmentation masks obtained through SAM's data engine. This dataset offers well-labeled, category-independent, high-quality, multi-granularity object masks. Utilizing these labeled masks enables the acquisition of necessary amodal annotations without human intervention via the proposed TAOS pipeline, as elaborated in Section 3.3.

**Amodal segmentation benchmarks.** To assess the effectiveness of our approach, we evaluate Amodal SAM on six amodal segmentation datasets including both image and video amodal segmentation benchmarks: KINS(Qi et al.), COCOA(Zhu et al.), COCOA-clsFollmann et al., MP3D-Amodal(Zhan et al., a), FISHBOWL(Tangemann et al., 2023), and MOViD-A(Gao et al.).

- KINS, an extensive traffic amodal dataset based on KITTI(Şenyiğit et al., 2014), comprises 14,991 images across 7 categories, with 7,474 images for training and the rest for testing.
- COCOA, a subset of the COCO dataset(Lin et al.), consists of 2,476 training images and 1,223 testing images.
- COCOA-cls is a dataset of objects selected from COCOA, containing 80 categories.
- MP3D-Amodal is constructed from MatterPort3D(Chang et al., 2017), where amodal masks are generated by projecting the 3D object structures onto the image.
- FISHBOWL is a video benchmark, captured from a publicly available WebGL demo of an aquarium.
- MOViD-A is a video-based synthesized dataset created from MOVi dataset.

**Evaluation protocols.** For evaluation, we employ the mean Intersection over Union (mIoU) as the primary metric, following (Gao et al.; Zhan et al., a). We compute both the ground-truth amodal mask ($\text{mIoU}_f$) and the occluded region ($\text{mIoU}_o$). The occluded mIoU evaluates the overall quality of the occluded section of the target object.

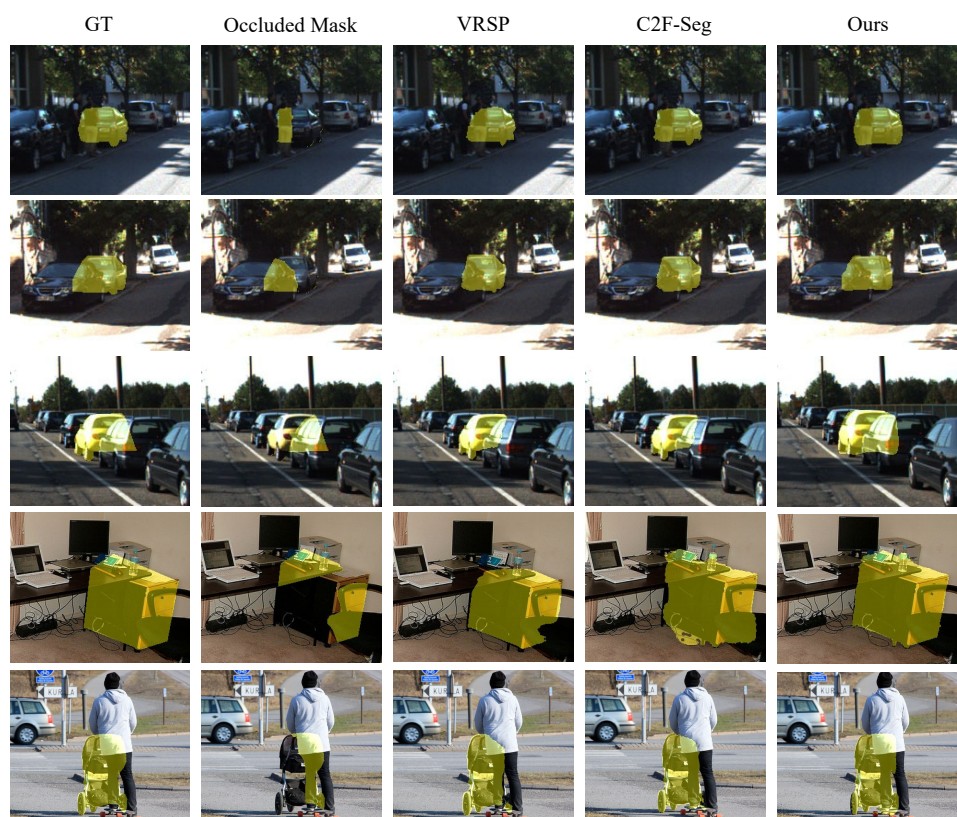

| GT | Occluded Mask | VRSP | C2F-Seg | Ours |

Figure 5: This figure presents the qualitative results of VRSP, C2F-Seg, and our method on KINS and COCOA datasets. "GT" represents the ground truth, and the "Occluded mask" indicates the occluded region of the target object.

**Implementation details.** We implement our method with PyTorch. In our experiments, we augment the bounding box with a mask as the model input, expanding the bounding box by a factor of 0.2 in all directions, resulting in an input area 1.44 times larger than the original. The iteration counts are set to 40k, 20k, 7k, 75k and 75k for KINS, COCOA, COCOA-cls, FISHBOWL and MOViD-A datasets, respectively. We utilize the AdamW optimizer with a learning rate that decreases from 1e-4 to 5e-5 throughout training.

Table 1: Performance comparison on three representative amodal datasets.

| METHOD | KINS | | COCOA | | COCOA-cls | |
|---|---|---|---|---|---|---|
| | $mIoU_f$ | $mIoU_o$ | $mIoU_f$ | $mIoU_o$ | $mIoU_f$ | $mIoU_o$ |
| PCNet(Zhan et al., b) | 78.02 | 38.14 | 76.91 | 20.34 | – | – |
| VRSP(Xiao et al.) | 80.70 | 47.33 | 78.98 | 22.92 | 79.93 | 26.72 |
| AISformer(Tran et al., b) | 81.53 | 48.54 | 72.69 | 13.75 | – | – |
| C2F-Seg(Gao et al.) | 82.22 | 53.60 | 80.28 | 27.71 | 81.71 | 36.70 |
| PLUG(Liu et al., 2024) | 88.10 | 61.42 | 83.23 | 32.88 | - | - |
| **Amodal SAM** | 88.79 | 63.12 | 84.27 | 59.94 | 87.65 | 54.34 |

## 4.2 EVALUATION OF AMODAL IMAGE SEGMENTATION

**Closed-domain amodal segmentation.** To have a comparison in the closed-domain scenario, *i.e.*, the training and testing samples are from the same dataset, we evaluate the proposed methods with four representative works on the KINS, COCOA, and COCOA-cls datasets. This comparison is presented in Table 1.

**Open-world amodal segmentation.**

To further assess the generalization capacity across different data domains, we compared our method with three representative works, including SDAmodal(Zhan et al., a), the pioneering work address-

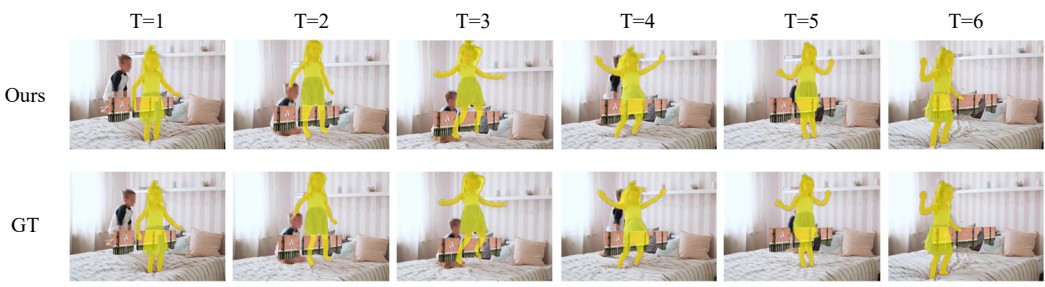

Figure 6: This figure illustrates the viability of transitioning our method to SAM-2. More examples and details are in the appendix A.3.2.

ing the open-world amodal segmentation task. For a fair comparison, we used the COCOA dataset as the training set and tested on the MP3D-Amodal dataset and the KINS dataset. The proposed method achieved better performance, especially in predicting occluded areas, as shown in Table 2.

**Qualitative evaluation.** The visual comparison is demonstrated in Figure 5, from which we can observe that the proposed model shows superior amodal segmentation ability compared to the recent state-of-the-art methods.

### 4.3 EVALUATION OF AMODAL VIDEO SEGMENTATION

For comparison in the video amodal segmentation, we conducted additional experiments using our model for the video amodal segmentation task. Table 3 presents a comparison between our approach and several competing methods on the FISHBOWL and MOViD-A datasets. Our Amodal SAM outperforms all current state-of-the-art techniques, including image-level and video-level baselines, across both datasets.

### 4.4 ABLATION STUDY

This section shows the ablation study conducted to assess the impact of our designs.

Table 2: Performance comparison across data domains.

| Method | MP3D-Amodal | | KINS | |
|---|---|---|---|---|
| | $mIoU_f$ | $mIoU_o$ | $mIoU_f$ | $mIoU_o$ |
| VRSP | 58.5 | 21.2 | 59.6 | 10.3 |
| C2F-Seg | 61.7 | 27.4 | 62.3 | 12.7 |
| SDAmodal | 76.4 | 38.5 | 76.0 | 17.9 |
| Amodal SAM | 78.2 | 62.1 | 77.1 | 50.2 |

Table 3: Performance comparison on video amodal segmentation datasets.

| Method | FISHBOWL | | MOViD-A | |
|---|---|---|---|---|
| | $mIoU_f$ | $mIoU_o$ | $mIoU_f$ | $mIoU_o$ |
| PCNET | 87.04 | 65.02 | 64.35 | 27.31 |
| AISformer | - | - | 67.72 | 33.65 |
| SaVos | 88.63 | 71.55 | 60.01 | 22.64 |
| C2F-Seg | 91.68 | 81.21 | 71.67 | 36.13 |
| Amodal SAM | 92.74 | 83.36 | 73.06 | 39.21 |

Table 4: Ablation study of the proposed designs.

| Method | COCOA | | COCOA-cls | |
|---|---|---|---|---|
| | $mIoU_f$ | $mIoU_o$ | $mIoU_f$ | $mIoU_o$ |
| EA + SCA | 73.78 | 33.39 | 82.70 | 33.61 |
| EA | 55.04 | 30.63 | 57.41 | 31.47 |
| SCA | 51.75 | 24.85 | 58.18 | 27.64 |

In all experiments, the model is trained on our customized amodal dataset obtained from SA-1B. In Table 4, "EA" denotes the Encoder-focused adaptation, and "SCA" denotes the spatial completion adapter. The 2nd row without "SCA" is implemented with the conventional adapter. The 3rd row without "EA" refers to the results obtained by inserting the adapters in the decoder. The comparison between the 1st and 3rd rows demonstrates that encoder-focused adaptation achieves better performance. By comparing the results of "EA+SCA" and "EA", we can observe that SCA yields much better results than the standard adapter. Further ablation study can be found in Appendix A.2.

### 5 CONCLUDING REMARKS

In this work, we presented Amodal SAM, a framework that successfully extends SAM's capabilities to open-world amodal segmentation through three synergistic aspects: model, data, and optimization. Our approach achieves state-of-the-art performance across multiple benchmarks while maintaining zero-shot capabilities, and can be readily extended to video applications. We believe this work can provide a solid foundation for future research in open-world visual scene understanding.

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

# A  APPENDIX

## OVERVIEW

This is the supplementary file for our submission titled *Amodal SAM: Open-World Amodal Segmentation*. This material supplements the main paper with the following content:

- (A.1) **Implementation Details**

  - (A.1.1) **Instruction of metrics**

  - (A.1.2) **Structure of the discriminator**

  - (A.1.3) **Structure of the SCA**

  - (A.1.4) **Adaptation to SAM-2**

- (A.2) **Additional Experiments**

  - (A.2.1) **Ablation study on the number of adapters**

  - (A.2.2) **Ablation study on the optimization adaptation**

  - (A.2.3) **Effectiveness of use points as prompt**

  - (A.2.4) **Effectiveness of pixel blurring to TAOS pipeline**

- (A.3) **Additional Visual Illustrations**

  - (A.3.1) **Image**

  - (A.3.2) **Video**

## A.1  DETAILS OF OUR METHODS

### A.1.1  INSTRUCTION OF METRICS

In Section 4.1 of the main paper, we introduce the evaluation metrics: $\text{mIoU}_{full}$ and $\text{mIoU}_{occ}$, following (Gao et al.; Zhan et al., a). $\text{mIoU}_{full}$ calculates the average Intersection over Union (IoU) between the predicted amodal masks (representing the entire object) and the ground-truth amodal

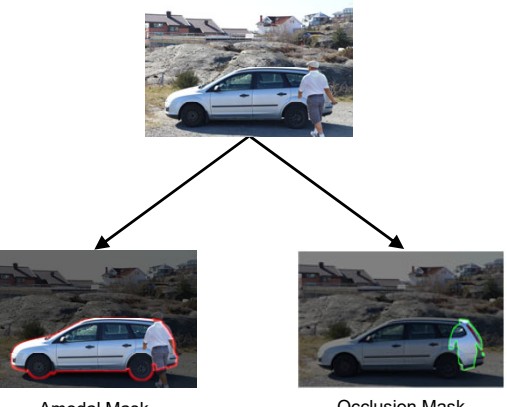

Figure 7: The occlusion mask delineates the occluded region within the amodal mask. $mIoU_f$ assesses the predictive accuracy of the amodal mask, whereas $mIoU_o$ evaluates the predictive accuracy of the occlusion mask. During evaluation, even if the model's predicted mask solely includes visible regions, $mIoU_f$ might surpass 50, whereas $mIoU_o$ could remain at 0.

masks. In contrast, $mIoU_{occ}$ measures the average IoU between the predicted occlusion masks (representing the occluded regions) and the ground-truth occlusion masks. It can be expressed as:

$$
\begin{aligned}
I_{occ} &= (M_{\text{pred}} - M_{\text{visible}}) \quad \& \quad (M_{\text{gt}} - M_{\text{visible}}) \\
U_{occ} &= (M_{\text{pred}} - M_{\text{visible}}) \quad | \quad (M_{\text{gt}} - M_{\text{visible}}) \\
mIoU_{occ} &= \frac{1}{n} \sum_{i=1}^{n} I_{occ}/U_{occ}
\end{aligned}
\tag{11}
$$

Where $M_{\text{pred}}$ denotes the predicted mask, $M_{\text{gt}}$ represents the ground truth mask, and $M_{\text{visible}}$ is the mask of the visible region. Figure 7 demonstrates the difference between amodal masks and occlusion masks. The $mIoU_{occ}$ metric specifically assesses the model's performance in predicting occluded regions, which is inherently more challenging than predicting visible regions in amodal segmentation. Therefore, $mIoU_{occ}$ is a crucial metric for evaluating model performance in this scenario.

### A.1.2 STRUCTURE OF THE DISCRIMINATOR

To effectively extend the capabilities of SAM from visible region segmentation to amodal segmentation while preserving its powerful zero-shot functionality, we propose Optimization Adaptation, as detailed in Section 3.4 of the main paper. This approach includes holistic topology regularization, which is based on adversarial learning and necessitates the employment of a discriminator for implementation.

Our Target-Aware Occlusion Synthesis (TAOS) technique enables the generation of occluded images while retaining the original unoccluded image. In the training of the open-domain model, the discriminators receive the original image and its corresponding mask as input. Specifically, the images and masks are concatenated, and their features are extracted using a multilayer network. These features undergo progressive downsampling, and the alignment between the predicted masks and the ground truth in terms of topology and morphology is assessed. To ensure robust consistency measurements, a sigmoid activation function is applied to confine the output values within a probability range of 0 to 1. During closed-domain training, where unobstructed images are not available, only the mask is utilized as input to the discriminator. In both training phases, the output from Amodal SAM serves as negative samples, while the ground truth acts as positive samples. This approach aims to align the distribution of the model-predicted mask with the ground truth, facilitating the learning of shape priors for achieving topological regularization as illustrated in Figure 8.

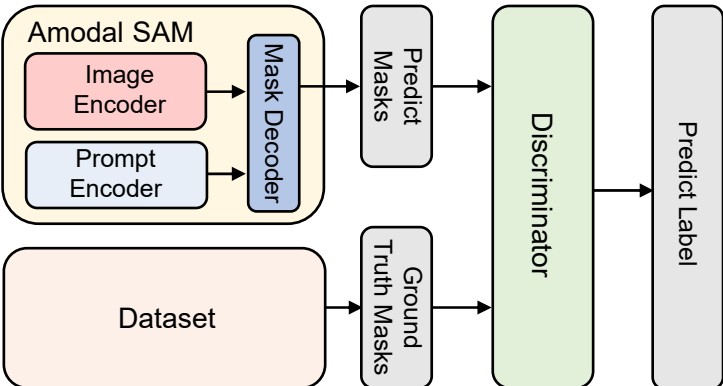

Figure 8: The discriminator uses the predictions from Amodal SAM as negative samples and the ground truth as positive samples. This alignment helps the model-predicted mask distribution closely match the ground truth, facilitating the learning of shape priors for achieving topological regularization.

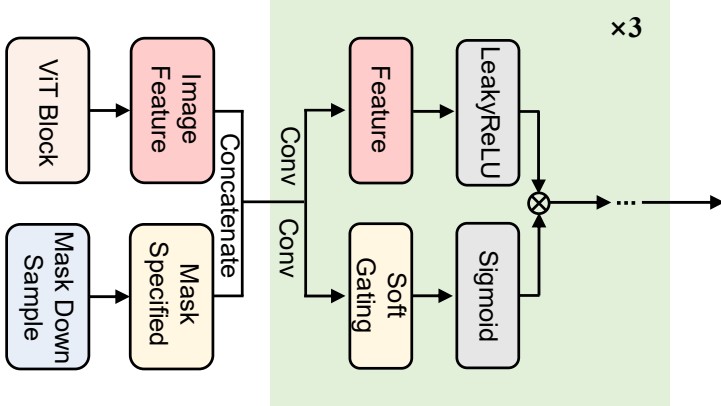

Figure 9: The Spatial Completion Adapter combines the image features and $M_{spec}$ by concatenating them as input, and then iterates the feature selection and completion process three times.

### A.1.3 STRUCTURE OF THE SCA

In the main paper, we introduce the Spatial Completion Adapter (SCA), which utilizes Gated Convolution. This approach incorporates dynamic feature selection to effectively leverage spatial information from $M_{\text{spec}}$, enabling the reconstruction of occluded regions through spatial soft gating in the feature space. Specifically, we concatenate the image features output by the preceding ViT Block with masks that have been downsampled to the same spatial resolution. Two distinct convolutional operations are then employed to extract features and perform soft gating, respectively. A sigmoid activation function is applied for soft gating to constrain the value range to [0,1], effectively approximating a switching mechanism to enhance feature selection. Concurrently, a LeakyReLU activation function is applied to the extracted features. The outputs of these two processes are subsequently multiplied to generate the final result as illustrated in Figure. This feature-completion operation is executed three times within a Spatial Completion Adapter (SCA). SCAs are integrated into the shallow, middle, and deep layers of the image encoder. In the additional experiment detailed in Section A.2.1, we will demonstrate the efficacy of incorporating three SCAs.

### A.1.4 ADAPTATION TO SAM-2

SAM-2 (Segment Anything Model 2) developed by Meta AI, is an image and video segmentation model. This model represents a significant advancement over its predecessor, SAM, as it excels in

segmenting diverse objects in both images and videos. SAM-2 empowers users to define a target object in any video frame or image using a point, box, or mask prompt. Subsequently, the model predicts the spatio-temporal segmentation mask for the target object across the entire video. Users can iteratively provide prompts on multiple frames for interactive refinement. Similar to SAM, SAM-2 comprises three core components:

- an image encoder, which extracts features from the image or each frame of the video.
- a prompt encoder, which processes user-provided prompts, such as points, boxes, or masks, to generate embeddings for target localization.
- a masking decoder, which integrates image features, prompt embeddings to produce a segmentation mask for the image or current frame.

In addition to these components, SAM-2 introduces specialized memory modules for video processing:

- a memory encoder, which fuses predicted masks with image encoder features to generate memory embeddings across frames.
- a memory attention layer, which employs cross-attention to link the current frame features with memory information, facilitating the integration of spatio-temporal context.

To adapt our method for SAM-2 and achieve amodal video segmentation, we follow the adaptation strategy applied to Amodal SAM. Specifically, we continue to use the Encoder-focused adaptation approach to incorporate our Spatial Completion Adapter(SCA) into the image encoder. Unlike SAM, SAM-2's image encoder employs a Hiera-based structure, which generates image features at four different spatial scales during the encoding process. The early-layer features, with larger spatial sizes, are used in the mask decoder to produce high-quality masks as described in, while the late-layer features, with smaller spatial sizes, are used to generate the final image embedding. Consequently, all four spatial-scale features contribute to the final mask generation.

To ensure that the spatial information across all four scales is fully enhanced, we integrate an SCA among the encoder blocks corresponding to each spatial scale. This allows us to spatially complete the image features at all four spatial scales, optimizing the segmentation process comprehensively as illustrated in Figure 10. Following SAM-2's original training methodology, we utilized both image and video datasets during the training process. Additionally, we incorporated the optimization strategies from Amodal SAM, including segmentation supervision, region consistency and holistic topological regularization, to enhance the training process.

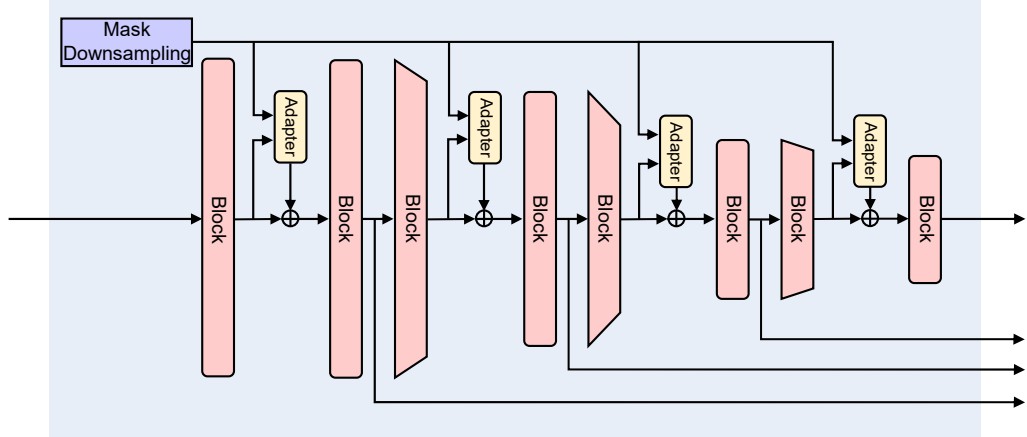

Figure 10: The image encoder of SAM-2 consists of four stages with distinct spatial scales. We incorporate an SCA at the outset of each stage to ensure that every feature output from the image encoder undergoes spatial completion.

## A.2 ADDITIONAL EXPERIMENTS

This section shows the additional experiments. In the first two experiments, our models were trained on the dataset's training set and evaluated on its corresponding testing set or validation set.

### A.2.1 ABLATION STUDY ON THE NUMBER OF ADAPTERS

As detailed in Section A.1.3, we integrated three SCAs into the image encoder. To assess the impact of varying the number of SCAs on the performance of Amodal SAM, we conducted experiments using different numbers of SCAs on the KINS and COCOA-cls datasets, with the results summarized in Table 5. The results demonstrate that our method of integrating SCAs achieves effective performance while minimally increasing the model's parameters, thereby maintaining efficient inference speed.

Table 5: Ablation study on the number of adapters

| Method | KINS | | COCOA-cls | |
|---|---|---|---|---|
| | $mIoU_f$ | $mIoU_o$ | $mIoU_{fu}$ | $mIoU_o$ |
| one adapter | 80.05 | 53.92 | 79.94 | 29.58 |
| two adapter | 81.26 | 56.34 | 80.74 | 32.97 |
| full model | 83.43 | 59.27 | 81.42 | 35.79 |

### A.2.2 ABLATION STUDY ON THE OPTIMIZATION ADAPTATION

In our optimization adaptation, we formulate a composite learning objective $\mathcal{L}$, encompassing not only the conventional $\mathcal{L}_{seg}$ but also regional consistency ($\mathcal{L}_{reg}$) and holistic topological regularization ($\mathcal{L}_{hol}$). The effectiveness of these newly introduced learning objectives, *i.e.*, $\mathcal{L}_{reg}$ and $\mathcal{L}_{hol}$, can be validated through the results presented in Table 6.

Table 6: Performance under different training objectives.

| Method | | KINS | | TAOS Dataset | |
|:---:|:---:|:---:|:---:|:---:|:---:|
| $\mathcal{L}_{\text{hol}}$ | $\mathcal{L}_{\text{reg}}$ | $\text{mIoU}_f$ | $\text{mIoU}_o$ | $\text{mIoU}_f$ | $\text{mIoU}_o$ |
| ✓ | ✓ | 88.79 | 63.12 | 90.85 | 54.64 |
| × | ✓ | 87.73 | 62.75 | 89.04 | 54.39 |
| ✓ | × | 88.07 | 61.94 | 89.97 | 53.52 |
| × | × | 87.02 | 60.73 | 88.45 | 53.41 |

### A.2.3 EFFECTIVENESS OF USE POINTS AS PROMPT

SAM features a versatile prompt encoder capable of handling various types of input prompts, such as points, boxes, and text. In the previous experiments, we used boxes as the prompt input. To evaluate whether Amodal SAM can also perform amodal segmentation with points as prompts, we designed this experiment. For comparison with the prior use of boxes, we randomly sampled two points within the visible region of the target object as the prompt input during both training and inference.

The experiments were conducted on the KINS and COCOA dataset, and the results are presented in Table 7. Experimental results show that amodal sam still achieves good performance using points as the prompt input. For the COCOA dataset, where object occlusion scenarios are more diverse and complex, using boxes as prompts may introduce ambiguity. In contrast, using points as prompts can mitigate this issue and even yield better performance.

Table 7: Performance comparison of different prompts.

| Prompt | KINS | | COCOA | |
|:---:|:---:|:---:|:---:|:---:|
| | $\text{mIoU}_f$ | $\text{mIoU}_o$ | $\text{mIoU}_f$ | $\text{mIoU}_o$ |
| Box | 83.43 | 59.27 | 82.54 | 51.66 |
| Points | 82.74 | 58.52 | 82.76 | 53.28 |

### A.2.4 EFFECTIVENESS OF PIXEL BLURRING TO TAOS PIPELINE

As detailed in Section 3.3 of the main paper, at the final stage of the TAOS-pipeline, pixel blurring is applied to the synthesized occlusion. Gaussian blur is applied after synthesizing occlusions to smooth the boundaries between the target and the occluded objects. This aims to prevent the model from overly focusing on jagged edges during training and instead encourage it to focus on the shapes of the objects. To validate the effectiveness of blurring on the model's generalization capability, we conducted an ablation experiment. In this experiment, we constructed two identical datasets except for the presence or absence of blurring. The model was trained separately on these two datasets and then tested on the COCOA-cls dataset with the results summarized in Table 8.

Table 8: Ablation study on the pixel blurring.

| Pixel Blurring | KINS | | COCOA-cls | |
|:---:|:---:|:---:|:---:|:---:|
| | $\text{mIoU}_f$ | $\text{mIoU}_o$ | $\text{mIoU}_f$ | $\text{mIoU}_o$ |
| ✓ | 73.79 | 43.72 | 82.13 | 33.56 |
| | 72.85 | 42.37 | 81.26 | 31.77 |

## A.3 ADDITIONAL VISUAL ILLUSTRATIONS

### A.3.1 IMAGE

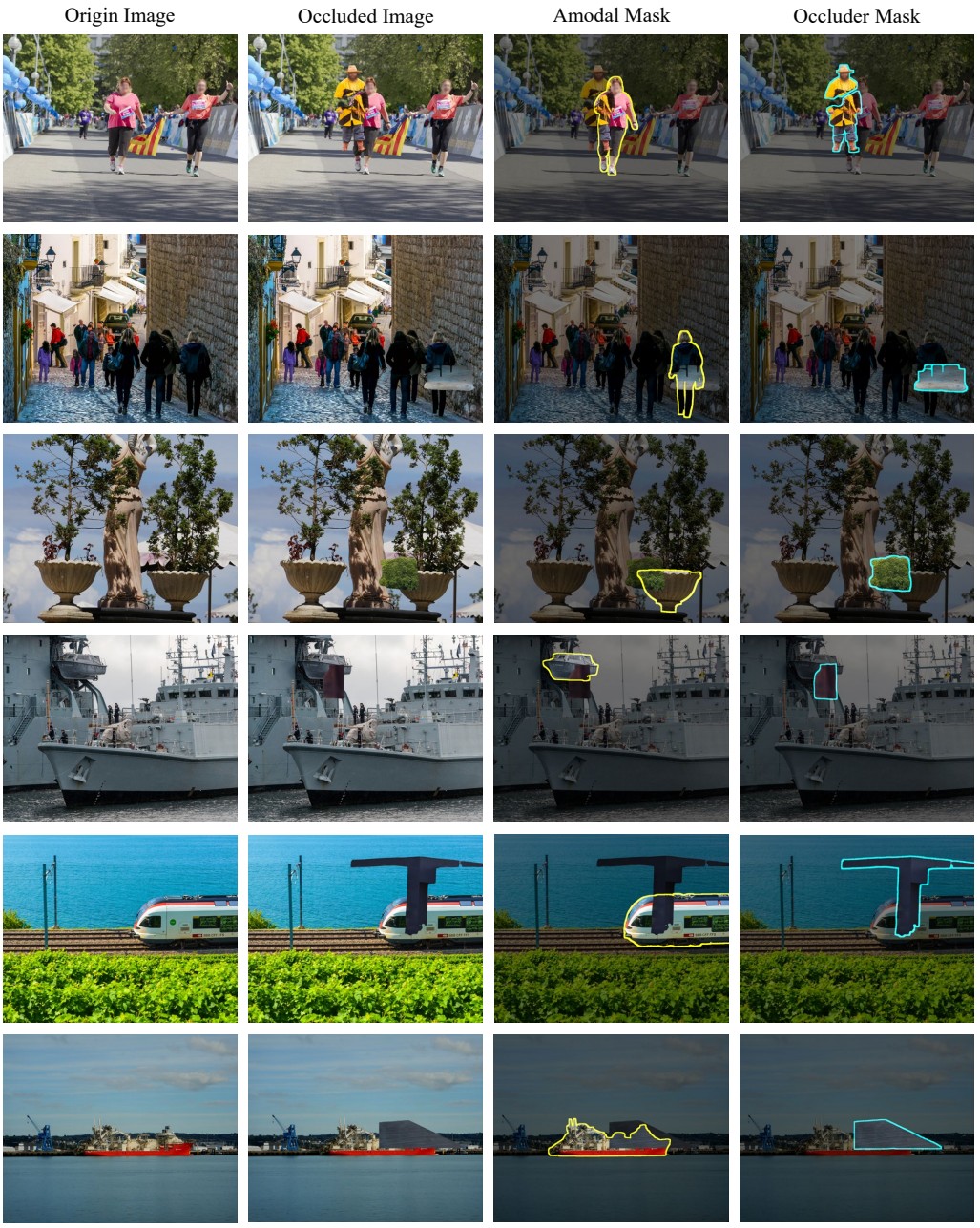

Figure 11: Examples of the datasets we constructed. For each example showing the original image, the image after the occlusion was added, the amodal mask, and the mask of the occluder respectively

Input Image                GT Mask                Predict Mask

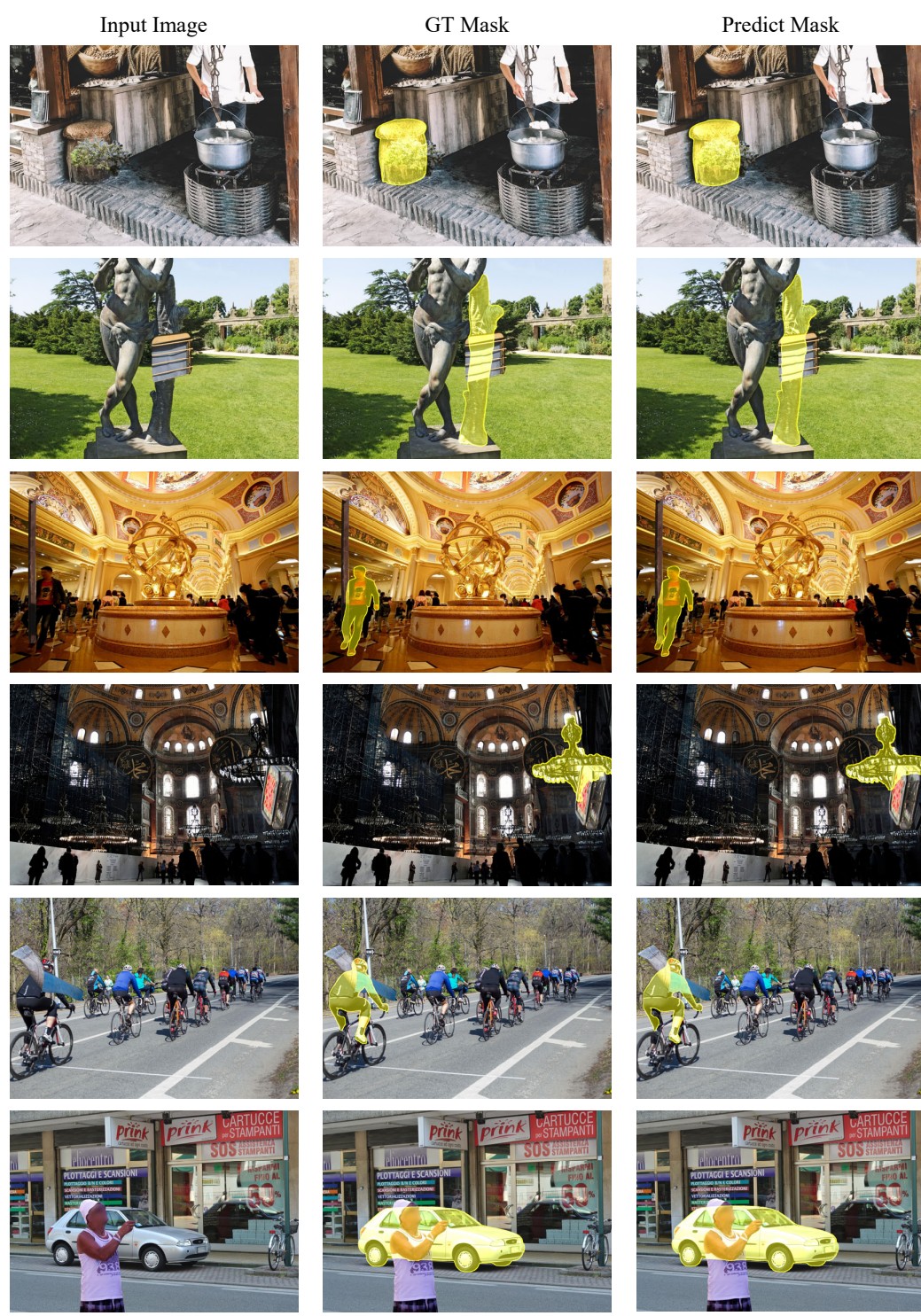

Figure 12: More qualitative results of Amodal SAM. "GT" represents the ground truth and the "Predict Mask" is the result of amodal sam prediction.

### A.3.2 VIDEO

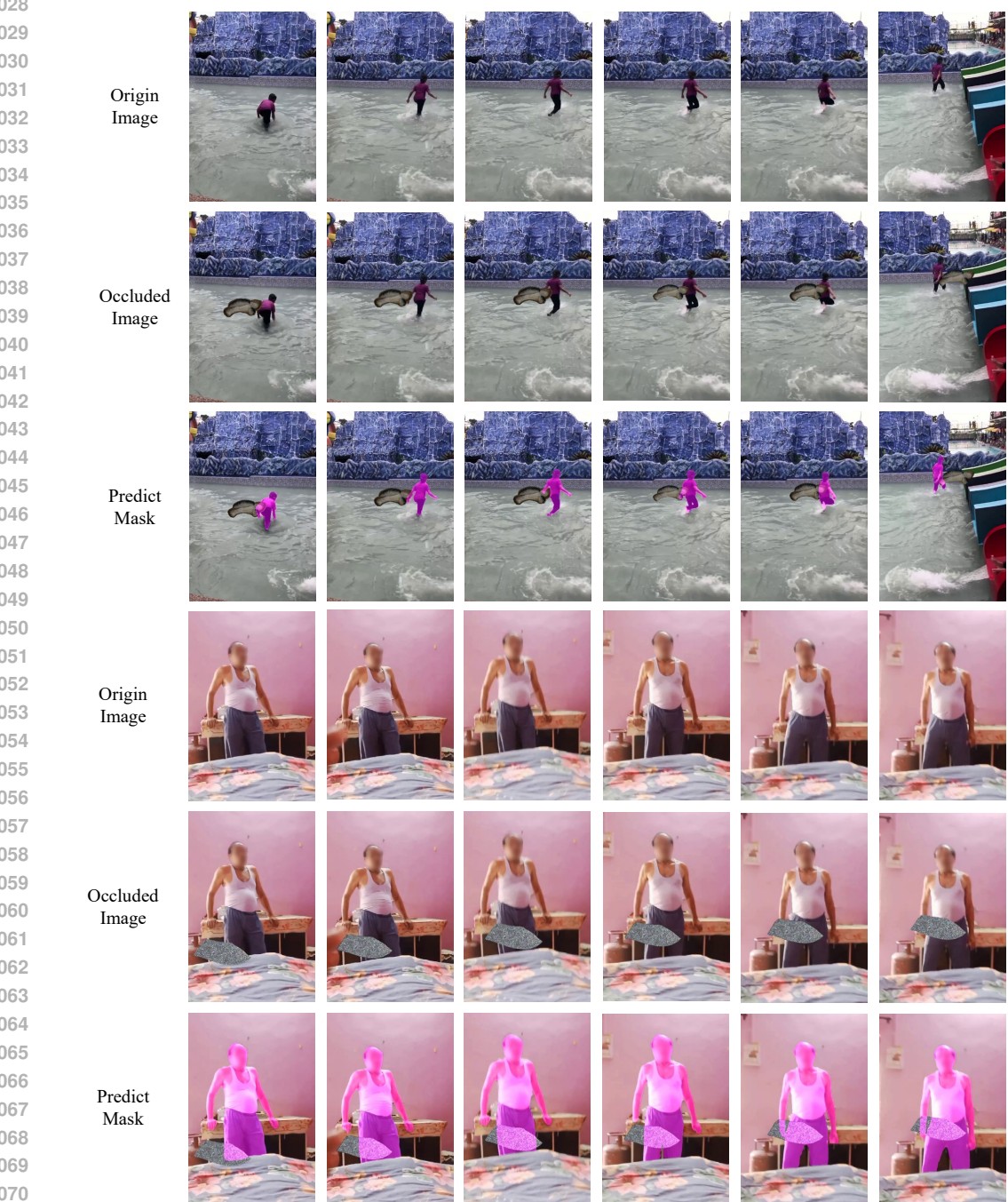

Figure 13: Example of Migrating Adaptation to SAM-2

