# OpenReview forum: "Amodal SAM: Open-World Amodal Segmentation"
_ICLR.cc/2026/Conference — ICLR 2026 Conference Withdrawn Submission_

### Official Review · Reviewer_5CQw · 2025-10-26

**Soundness:** 1
**Presentation:** 2
**Contribution:** 2
**Rating:** 2
**Confidence:** 4

**Summary:**

The submission focused on the task of amoda segmentation, which predict the object shapes including occluded regions. Specifically, the paper proposed to extend SAM to a amodal SAM with several proposed modules. The proposed Spatial Completion Adapter reconstructed the occluded regions, the Target-Aware Occlusion Synthesis pipeline generated training data, and the proposed training loss enhance the consistency on regional and topological aspects. Extensive experiments on various datasets are provided to show the effectiveness of the proposed method.

**Strengths:**

1. The task of open world amodal segmentation is interesting.

2. The idea of building amodal segmentor based on SAM is reasonable and make sense.

3. The performances of proposed method are shown on various benchmark datasets, and outperform baselines by a large margin.

**Weaknesses:**

1. The insight of proposed regional consistency is problematic. Why are the visible and occluded regions of the same object expected to exhibit similar characteristics like appearance and texture patterns? The occluded regions should be in the characteristics of occluder, instead of target. This is a factual and fatal issue.

2. The critical details of proposed TAOS is unclear. How to employ VLM to evaluate the generated occlusion and eliminate invalid data? How to promise that the generated amodal mask is complete, if the selected target is already occluded before synthesis?

3. The critical details of proposed holistic topological regularization is unclear. How to compute the Eqn. 10? The detailed formulations are also missing in A.1.2.

4. The experiments is in-depth enough. How to balance the loss terms in Eqn. 7? Why is there no hyperparameter?

5. The performances seem not cost-effective enough against previous works, bencause the proposed method relies on a large number of data and a heavy foundation model. Related analysis on complexity is also missing.

6. Some closely related works on amodal segmentation are missing, and thus the discussion and experiments are not extensive enough.&#x20;

   > 1. "pix2gestalt: Amodal segmentation by synthesizing wholes." 2024 IEEE/CVF Conference on Computer Vision and Pattern Recognition (CVPR). IEEE Computer Society, 2024.
   >
   > 2. "Amodal segmentation through out-of-task and out-of-distribution generalization with a bayesian model." Proceedings of the IEEE/CVF conference on computer vision and pattern recognition. 2022.
   >
   > 3. "Amodal instance segmentation via prior-guided expansion." Proceedings of the AAAI Conference on Artificial Intelligence. Vol. 37. No. 1. 2023.
   >
   > 4. "Amodal cityscapes: a new dataset, its generation, and an amodal semantic segmentation challenge baseline." 2022 IEEE Intelligent Vehicles Symposium (IV). IEEE, 2022.

**Questions:**

Please see Weakness.

---

### Official Review · Reviewer_QvEt · 2025-10-26

**Soundness:** 3
**Presentation:** 3
**Contribution:** 1
**Rating:** 2
**Confidence:** 4

**Summary:**

The paper proposes Amodal SAM, a framework that extends SAM to predict both visible and occluded regions of objects, aiming to improve generalization in open-world amodal segmentation. The authors introduce a lightweight Spatial Completion Adapter to infer hidden regions, a Target-Aware Occlusion Synthesis pipeline that generates synthetic occlusions from the SA-1B dataset without manual labeling, and new loss terms promoting regional consistency and topological coherence. Evaluations on six benchmarks spanning both image and video amodal segmentation (KINS, COCOA, COCOA-cls, MP3D-Amodal, FISHBOWL, and MOViD-A) show consistent gains over several prior methods in both closed- and cross-domain settings.

While the method demonstrates solid empirical performance, its core ideas substantially overlap with prior work. The proposed TAOS pipeline closely parallels existing synthetic occlusion generation strategies such as Amodal-LVIS in SAMEO and the mixed real–synthetic approach in SAMBA, both of which already integrate similar data synthesis procedures into SAM-based models (neither is cited). Moreover, the paper omits direct comparisons to these closely related and publicly available baselines — pix2gestalt, SAMEO, and SAMBA — which weakens the empirical validation and makes it difficult to substantiate claims of state-of-the-art performance or novel methodological contribution.

**Strengths:**

The paper is relatively well written and easy to follow.

The proposed approach is sound.

The proposed design offers a natural extension from image to video amodal segmntation.

A minimal ablation study is reported.

**Weaknesses:**

The authors completely omit the most relevant works in the literature on open-world amodal segmntation. Specifically, they do not cite, discuss or compare to pix2gestalt [1], SAMEO [2], and SAMBA [3].

Moreover the dataset and methodological contributions are minimal compared to SAMEO and SAMBA, which also fine-tune SAM for amodal segmntation using synthetically "generated" occlusions.

[1] Ozguroglu, E., Liu, R., Surís, D., Chen, D., Dave, A., Tokmakov, P., and Vondrick, C. “pix2gestalt: Amodal Segmentation by Synthesizing Wholes.”, CVPR'24

[2] Tai, W.-E., Shih, Y.-L., Sun, C., Wang, Y.-C. F., and Chen, H.-T. “Segment Anything, Even Occluded.”, CVPR'25

[3] Liu, Z., Qiao, L., Chu, X., Ma, L., and Jiang, T. “Towards Efficient Foundation Model for Zero-shot Amodal Segmentation.”, CVPR'25

**Questions:**

Please discuss your work's relationship to the state-of-the-art methods for the problem you are trying to address. Update novelty claims accordingly.

Compare to these methods on the datasets used in their paper using the same metrics.

---

### Official Review · Reviewer_HrqS · 2025-10-30

**Soundness:** 2
**Presentation:** 2
**Contribution:** 2
**Rating:** 4
**Confidence:** 4

**Summary:**

This paper addresses the challenge of open-world amodal segmentation, where models need to predict complete object shapes including occluded regions and generalize to novel objects and contexts.
The authors propose Amodal SAM, a framework that extends SAM’s capabilities to amodal segmentation while preserving its generalization ability. This framework improves across three aspects: a lightweight Spatial Completion Adapter that enables the model to reconstruct occluded regions, a Target-Aware Occlusion Synthesis pipeline that generates diverse synthetic amodal training data to solve the scarcity of amodal annotations, and novel learning objectives that enforce regional consistency and holistic topological regularization. Extensive experiments show Amodal SAM achieves state-of-the-art performance on multiple image and video amodal segmentation benchmarks.

**Strengths:**

1. The paper is well-structured and clearly written, ensuring good readability.

2. Video Extension: As the first attempt to tackle open-world video amodal segmentation via SAM, the work successfully extends the method’s applicability beyond images and validates its generalization.

3. The authors provide comprehensive experimental validation.

**Weaknesses:**

1. Lack of a Related Work Section: Relevant studies are only scattered in the Introduction and experimental comparisons. A systematic review is missing, making it difficult to clearly position the work’s incremental contributions against existing literature.

2. Insufficient Investigation of Prior Works: When discussing video amodal segmentation, it overlooks the first work that addressed this task and validated it on open-world datasets[1]. This omission weakens the paper’s ability to demonstrate the novelty of its own video amodal segmentation efforts.

3. Unspecified Sources for Baselines in Table 3. The baseline methods included in Table 3 do not have corresponding literature sources.

4. Although Appendix A.2.1 verifies the impact of the number of Spatial Completion Adapters on performance, it does not conduct in-depth exploration in two key aspects: (1) how the insertion positions of SCA influence the model’s ability to reconstruct occluded regions; (2) comparative experiments between SCA and other mainstream adapter structures in the context of amodal segmentation tasks.

[1] Self-supervised Amodal Video Object Segmentation.

**Questions:**

1. How does the model perform on the open-world dataset mentioned in [1]?

2. How were the experimental results of each baseline method in Table 3 obtained?

[1] Self-supervised Amodal Video Object Segmentation.

---

### Official Review · Reviewer_BqqU · 2025-10-31

**Soundness:** 4
**Presentation:** 3
**Contribution:** 4
**Rating:** 6
**Confidence:** 3

**Summary:**

Amodal SAM extends the Segment Anything Model to perform open-world amodal segmentation, predicting both visible and occluded object regions even for unseen categories. It introduces a Spatial Completion Adapter for occlusion reasoning, a Target-Aware Occlusion Synthesis pipeline for large-scale training without manual labels, and new consistency losses for structural coherence. The model achieves state-of-the-art performance on multiple benchmarks, making it a strong framework for enabling amodal segmentation in open-world scenarios.

**Strengths:**

The paper extends SAM to perform amodal segmentation, predicting both visible and occluded object regions. It introduces a Spatial Completion Adapter (SCA) with new losses and leverages the TAOS synthetic dataset for training, showing strong technical quality.

**Weaknesses:**

There’s a typo in Figure 2: it labels the component as “Amodal Spatial Attention Module (SAM)”, which is confusing.

**Questions:**

1. Which components or models were actually retrained or fine-tuned during the development of Amodal-SAM? For example, was the base SAM frozen while only the Spatial Completion Adapter (SCA) was trained, or were other parts of the model updated as well?

2. After fine-tuning with the Spatial Completion Adapter (SCA) for amodal segmentation, does the model maintain the original SAM’s segmentation capability on normal (visible-only) tasks, or is there any performance degradation in standard segmentation scenarios?

---

### Note · Authors · 2025-11-13

I have read and agree with the venue's withdrawal policy on behalf of myself and my co-authors.